# Effective mechanical potential of cell–cell interaction explains three-dimensional morphologies during early embryogenesis

Hiroshi Koyama[1,2]*, Hisashi Okumura[2,3,4], Atsushi M. Ito[2,5], Kazuyuki Nakamura[6,7], Tetsuhisa Otani[2,8], Kagayaki Kato[2,9,10], Toshihiko Fujimori[1,2]

**1** Division of Embryology, National Institute for Basic Biology, Myodaiji, Okazaki, Aichi, Japan, **2** SOKENDAI (The Graduate University for Advanced Studies), Hayama, Kanagawa, Japan, **3** Biomolecular Dynamics Simulation Group, Exploratory Research Center on Life and Living Systems (ExCELLS), National Institutes of Natural Sciences, Myodaiji, Okazaki, Aichi, Japan, **4** Institute for Molecular Science, National Institutes of Natural Sciences, Myodaiji, Okazaki, Aichi, Japan, **5** National Institute for Fusion Science, National Institutes of Natural Sciences, Toki, Gifu, Japan, **6** School of Interdisciplinary Mathematical Sciences, Meiji University, Nakano-ku, Tokyo, Japan, **7** JST, PRESTO, Kawaguchi, Saitama, Japan, **8** Division of Cell Structure, National Institute for Physiological Sciences, Myodaiji, Okazaki, Aichi, Japan, **9** Bioimage Informatics Group, Exploratory Research Center on Life and Living Systems (ExCELLS), National Institutes of Natural Sciences, Myodaiji, Okazaki, Aichi, Japan, **10** Laboratory of Biological Diversity, National Institute for Basic Biology, Myodaiji, Okazaki, Aichi, Japan

* hkoyama@nibb.ac.jp

**Data Availability Statement:** The data used to produce the results and the resultant data in this manuscript are available from https://doi.org/10. 6084/m9.figshare.21714842. The source codes in

## Abstract

Mechanical forces are critical for the emergence of diverse three-dimensional morphologies of multicellular systems. However, it remains unclear what kind of mechanical parameters at cellular level substantially contribute to tissue morphologies. This is largely due to technical limitations of live measurements of cellular forces. Here we developed a framework for inferring and modeling mechanical forces of cell–cell interactions. First, by analogy to coarse-grained models in molecular and colloidal sciences, we approximated cells as particles, where mean forces (i.e. effective forces) of pairwise cell–cell interactions are considered. Then, the forces were statistically inferred by fitting the mathematical model to cell tracking data. This method was validated by using synthetic cell tracking data resembling various *in vivo* situations. Application of our method to the cells in the early embryos of mice and the nematode *Caenorhabditis elegans* revealed that cell–cell interaction forces can be written as a pairwise potential energy in a manner dependent on cell–cell distances. Importantly, the profiles of the pairwise potentials were quantitatively different among species and embryonic stages, and the quantitative differences correctly described the differences of their morphological features such as spherical vs. distorted cell aggregates, and tightly vs. non-tightly assembled aggregates. We conclude that the effective pairwise potential of cell–cell interactions is a live measurable parameter whose quantitative differences can be a parameter describing three-dimensional tissue morphologies.

this manuscript are uploaded on gitHub with https://doi.org/10.5281/zenodo.7427050.

**Funding:** This work was supported by following grants: Japan Ministry of Education, Culture, Sports, Science and Technology Grant-in-Aid for Scientific Research on Innovative Areas "Cross-talk between moving cells and microenvironment as a basis of emerging order" for H.K., the National Institutes of Natural Sciences (NINS) program for cross-disciplinary science study for H.K., a Japan Society for the Promotion of Science (JSPS) Grant-in-Aid for Young Scientists (B) for H.K. (17K15131) and a Japan Society for the Promotion of Science (JSPS) Grant-in-Aid for Transformative Research Areas (A) for T.F. (22H05168). The funders had no role in study design, data collection and analysis, decision to publish, or preparation of the manuscript.

**Competing interests:** The authors have declared that no competing interests exist.

## Author summary

Emergence of diverse three-dimensional morphologies of multicellular organisms is one of the most intriguing phenomena in nature. Due to the complex situations in living systems (e.g. a lot of genes are involved in morphogenesis.), a model for describing the emergent properties of multicellular systems has not been established. To approach this issue, approximation of the complex situations to limited numbers of parameters is required. Here, we searched for mechanical parameters for describing morphologies. We developed a statistical method for inferring mechanical potential energy of cell–cell interactions in three-dimensional tissues; the mechanical potential is an approximation of various mechanical components such as cell–cell adhesive forces, cell surface tensions, etc. Then, we showed that the quantitative differences in the potential is sufficient to reproduce basic three-dimensional morphologies observed during the mouse and *C. elegans* early embryogenesis, revealing a direct link between cellular level mechanical parameters and three-dimensional morphologies. Our framework provides a noninvasive tool for measuring spatiotemporal cellular forces, which would be useful for studying morphogenesis of larger tissues including organs and their regenerative therapy.

## Introduction

In multicellular living systems, various three-dimensional morphologies are observed in tissues and organs, which are often tightly linked to their physiological functions. Morphogenetic events are thought to be primarily dependent on the mechanical properties of the constituent cells [1–5]. Mechanical parameters such as cell–cell adhesion energy and cell surface tensions are involved in morphogenetic events including compartmentalization of different cell populations, epithelial cell movements etc. However, mechanical parameters which give rise to a variety of three-dimensional morphologies remain to be elucidated. If such parameters are identified, we can understand not only mechanical principles of emergent properties of multicellular systems during morphogenesis but also easily simulate three-dimensional tissues in a predictable manner, which may be valuable for regenerative therapy.

To search for mechanical parameters which are directly linked to three-dimensional morphologies, we focused on mechanical potential energies of pairwise cell–cell interactions as follows. Cell–cell adhesion forces, which are mediated by cell adhesion molecules such as cadherin proteins, can be approximated as attractive forces in isolated two cell systems (Fig 1A-i and 1A-ii), whereas both excluded volume effect of cells and actomyosin-mediated cell surface tensions can be as repulsive forces [1,2,4,6–8]; due to cell volume conservation, two cells cannot extremely approach each other, and the cell surface tensions can antagonize cell–cell adhesion forces. Because the summation of these forces would be dependent on cell–cell distance (Fig 1A-iii), it is written by pairwise potential energy (Fig 1A-iv, distance–potential curve). There may be factors other than the above affecting the pairwise potentials. In molecular and colloidal sciences, pairwise potentials of objects such as amino acids are considered, and the profiles of the distance–potential curves are critical for the behaviors of systems [9,10]. In multicellular systems, the pairwise potentials are conceptually well-known [7,8,11,12]. But the profiles have been never measured nor inferred in multicellular systems, and therefore, researchers arbitrarily set the profiles in theoretical studies (e.g. hard- or soft-core potentials, etc.) [13–16]. Theoretical studies related to morphogenesis did not assume substantial differences in the profiles [17–20]. However, considering the case of molecular/colloidal sciences, we speculated that the profiles substantially determine the systems' behaviors or morphologies in certain multicellular systems.

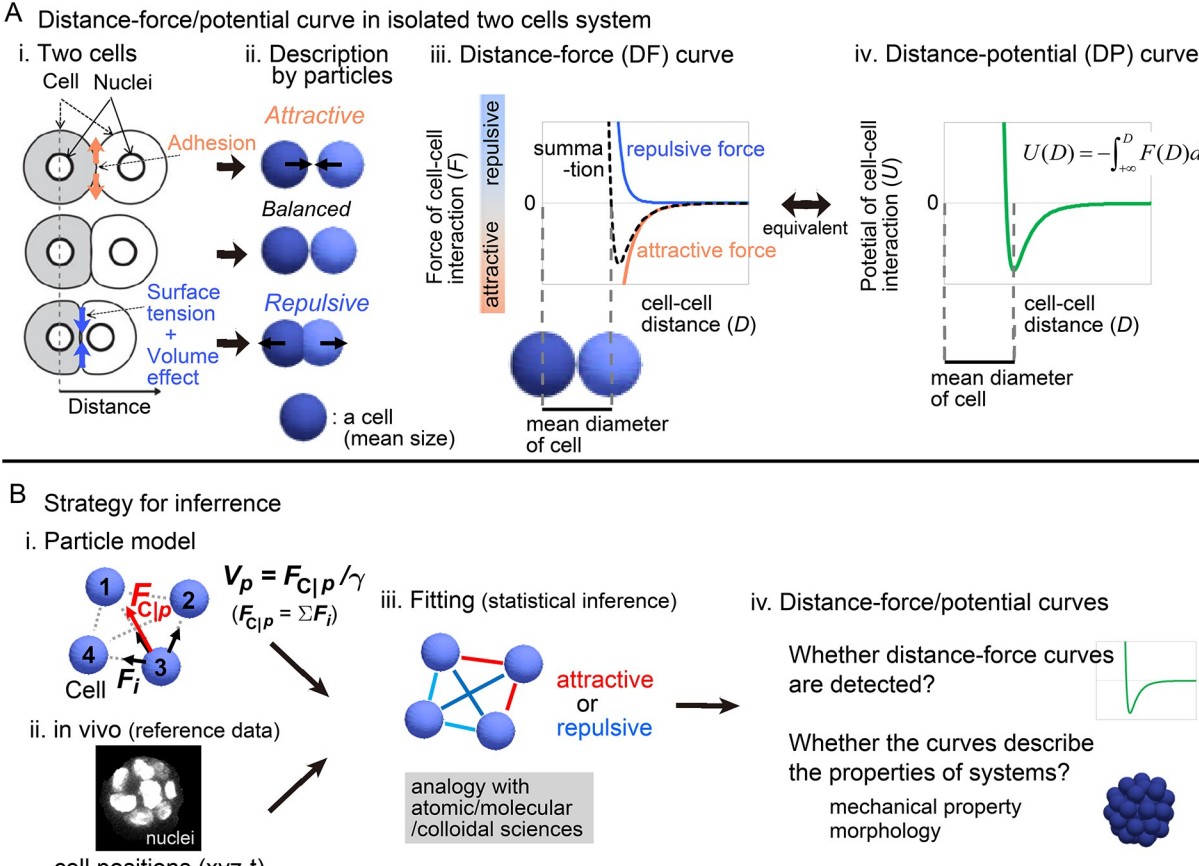

**Fig 1. Overview of strategy for inferring the effective potential of cell–cell interactions.** A. Relationship between microscopic forces and attractive/repulsive forces in isolated two cell systems. i) Microscopic forces are exemplified. ii) The microscopic forces are approximated as attractive/repulsive forces where cells are described as particles with the mean size. iii) Both the repulsive and attractive forces are provided as cell–cell distance–dependent functions, and the summation (black broken line) is the distance–force curve of the two particles. The relationship between the curve and the mean diameters of the cells is shown. iv) A distance–potential curve is shown where the mean diameters of cells correspond to the distance at the potential minimum. B. Strategy for inferring effective forces of cell–cell interactions. i) Particle model. A blue sphere corresponds to a cell. Attractive or repulsive force between the particles (blue spheres #1–4), was considered; vectors of cell–cell interaction forces ($F_i$) are illustrated by black arrows in the case of particle #3. The net force ($F_{C|p}$) of particle #3 is shown by a red vector. The summation of $F_i$ results in $F_{C|p}$ which determines the velocity ($V_{C|p}$) of $p$th particle. ii) Nuclear tracking data were obtained and used as a reference data during the inference. iii) Effective forces of cell–cell interactions were inferred by fitting. Red line, attractive; blue line, repulsive. iv) From the inferred effective forces, we examined whether distance–force/potential curves are detected. Related figure: S1 Fig (inference method).

To examine whether the profiles of pairwise potentials can determine morphologies of multicellular systems, quantitative measurements/inferences of the profiles in real tissues are inevitable. In the case of isolated two cell systems, adhesive forces between two cells were experimentally measured (i.e. forces required to dissociate one cell from the other) [4,21], but their pairwise potentials were not determined. Importantly, due to complex situations in real tissues compared with isolated two cell systems, it is unknown whether pairwise potentials are correctly defined and detectable *in vivo*. On the other hand, in molecular and colloidal sciences, we found a strategy worth considering: a top-down approach is adopted for inferring pairwise potentials, where positions of the objects such as radial distribution functions are solely used [10,22].

In the present study, we develop a top-down method for inferring pairwise potentials of cell-cell interactions by using movements of cells obtained from three-dimensional time lapse imaging (Fig 1B). Briefly, a theoretical model considering force values of cell–cell interactions were fitted to the cell tracking data. Our method was validated by using various synthetic data which were generated by simulations under pregiven cell–cell interaction forces. Then, we applied our method to the blastomeres in the *C. elegans* and mouse early embryos, and successfully detected pairwise potentials of cell–cell interactions. We discovered quantitative differences in the profiles of the inferred potentials among the embryos, and showed that, through simulations, the differences were linked to the embryos' morphological features. Note that applicability of our method to other cell types such as epithelial cells and self-migratory cells is beyond the scope of the present study. We assume that the nearly-spherical cell shape of blastomeres may be well-approximated by particle models, while the highly deformed cell shapes of epithelial or mesenchymal cells may not be suitable. This issue will be discussed later in the Discussion section with the limitations of our method.

## Theory and principle for inferring effective potential

### Particle-based cell models

To infer effective forces of cell–cell interactions *in vivo*, we developed a particle-based model in which the particles interact with each other and attractive or repulsive forces ($F_i$) are assumed as shown in Fig 1B-i; $i$ is an identifier for particle–particle interactions. In three-dimensional cellular systems with no attachment to substrates, we did not assume persistent random walks which originate from cellular traction forces on substrates [23,24].

The equation of particle motions is defined below. In cellular-level phenomena, viscous drag force or frictional force provided by the surrounding medium or tissue is dominant, as expected from the low Reynolds number; consequently, the inertial force is negligible in general [3,7,25–27]. The forces in such a system can be assumed to be correlated with the velocities of the objects [3,14,25,28,29]. Thus, the velocity ($V_{C|p}$) of a particle is calculated by the net force ($F_{C|p}$) exerted on the particle as follows (Fig 1B-i):

$$V_{C|p} = F_{C|p}/\gamma \tag{1}$$

where $p$ is an identifier for particles and $\gamma$ is the coefficient of viscous drag and frictional forces. $F_{C|p}$ is the summation of cell–cell interaction forces exerted on the $p$th particle. We assumed the simplest situation, i.e., $\gamma$ is constant (= 1.0). Thus, by giving the values of $F_i$ from, for instance, a distance–potential curve, a simulation can run. In addition, $\gamma$ may not be constant in real tissues, while it is challenging to measure the value with spatial resolution at cellular level [30].

### Data acquisition of time series of cell positions

To define cell positions, we focused on nuclei, because they are most easily imaged by microscopy (Fig 1B-ii) in a wide range of organisms from *C. elegans* to mammals, and these data are accumulating [31–35]. We utilized publicly available nuclear tracking data of a developing embryo of *C. elegans* [31]; nuclear tracking data of developing mouse embryos were obtained in this study.

### Development of method for inferring effective potentials/forces of cell-cell interaction

In the case of ions, molecules, etc., radial distribution functions are often used to infer the effective potentials of their interactions [10,22]. This method is simple, but is only applicable to thermodynamically equilibrium systems because this function reflects the effective

potentials under equilibrium states. Therefore, this method is not suitable for non-equilibrium systems, including morphogenetic events. To infer the effective potentials in non-equilibrium systems, we fitted the particle-based cell model to the time series of nuclear positions, where the fitting parameters are forces between all pairs of cell–cell interactions for each time frame (Fig 1B-iii). We systematically searched for the values of the effective forces between all pairs of cell–cell interactions that minimized the differences (i.e. corresponding to least squares) between the particle positions in the simulations and the *in vivo* nuclear positions. The differences ($G_{xyz}$) to be minimized are defined as follows.

$$G_{xyz} = \sum_{t=1}^{T} \sum_{p=1}^{P(t)} \frac{\{x_p(t) - x_p^{\text{ref}}(t)\}^2 + \{y_p(t) - y_p^{\text{ref}}(t)\}^2 + \{z_p(t) - z_p^{\text{ref}}(t)\}^2}{\Delta t} \qquad (2)$$

Here, $p$ is an identifier for particles, $t$ is an identifier for time frames, and $\Delta t$ is the time interval between the time frames. $T$ is the total time frame, and $P(t)$ is the number of particles at each time frame. The $x$, $y$, and $z$ coordinates of the $p$th particle obtained from microscopic images are $x_p^{\text{ref}}$, $y_p^{\text{ref}}$, and $z_p^{\text{ref}}$; ref means reference. The coordinates of the simulations are $x_p$, $y_p$, and $z_p$. Note that, because $\gamma$ was assumed to be constant in Eq 1, we can only infer relative but not absolute values of effective forces.

To determine whether our method can correctly infer effective forces, we applied it to synthetic data generated by simulations under given potentials of particle–particle interactions, and examined whether the inferred effective forces were consistent with the given potentials. Unfortunately, Eq 2 did not work well: the inferred forces were not consistent with given potentials. In particular, even around the longer cell–cell distance, the force values did not decay (S2 Fig). Therefore, we tried to incorporate various additional constraints into Eq 2. We found that the following cost function worked well: the function was set so that force values approach zero at long distant regions of cell–cell interactions as defined below:

$$G = G_{xyz} + G_{F_0} \left( G_{F_0} = \omega_{F_0} \sum_{t=1}^{T} \sum_{i=1}^{I(t)} \psi(D_i) F_i^2 \Delta t \right) \qquad (3)$$

where $F_i$ is the force value of $i$th cell–cell interactions, $I(t)$ is the total number of the cell–cell interactions for each time frame, $D_i$ is the cell–cell distance of the $i$th interaction, and $\omega_{F_0}$ is a coefficient for determining the relative contribution of the second term to $G$. $\Psi(D_i)$ is a distance-dependent weight that exponentially increases and eventually becomes extremely large, leading to that the value of $F_i$ approaches zero especially around larger $D_i$ (S1A-iii Fig). We show validations of this cost function in the next sections. Detailed procedures are described in S1 Text (Section 4).

## Results

### Systematic validation of inference method using simulation data

We systematically validated our inference method using simulation data. We used the Lenard–Jones (LJ) potential as a test case of the given potentials (Fig 2A-i), and generated simulation data under LJ potential, to which we applied our inference method. Importantly, we assumed cell-intrinsic mechanical activities that were introduced as fluctuations of the forces relative to the LJ potential (Fig 2A-i, "fluctuation (% of LJ)"), because the cells rapidly stop movements without such activities.

In Fig 2A-i, the particles were initially scattered to some extent, and then assembled. The inferred effective forces for each pair of particles for each time frame were plotted against

## A   Assembling process

### i. Simulation

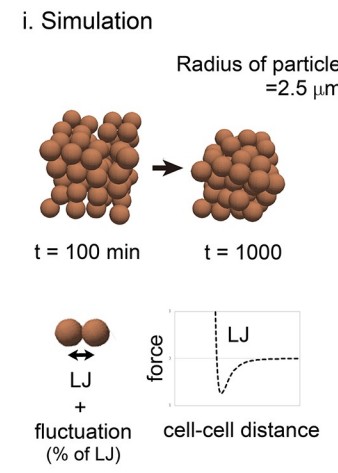

### ii. Plot & binned average of inferred effective force

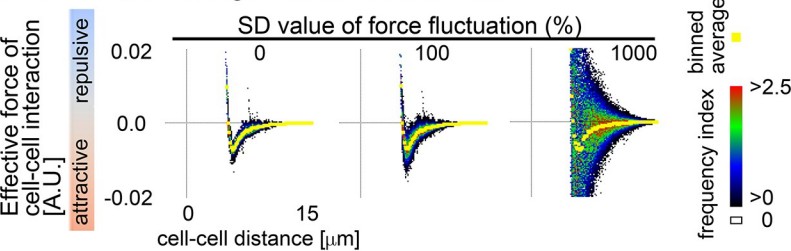

### iii. Inferred effective distance-force curve

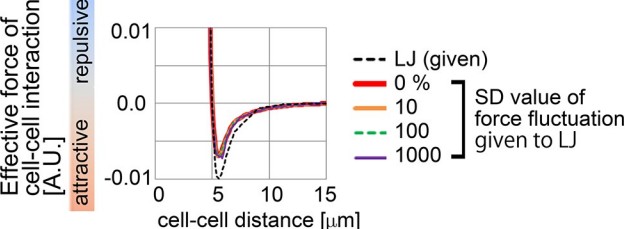

## B   Steady state

### i. Simulation

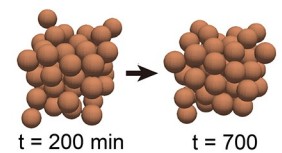

### ii. Inferred effective distance-force curve

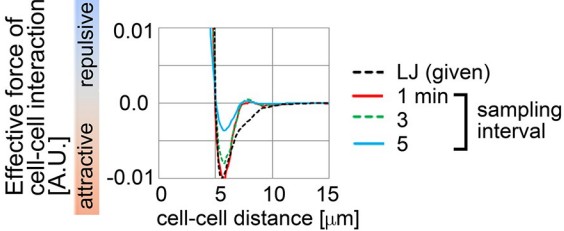

## C   Cell proliferation

### i. Simulation

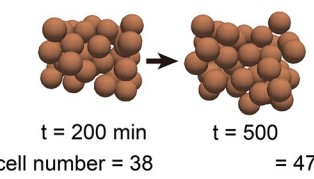

### ii. Inferred effective distance-force curve

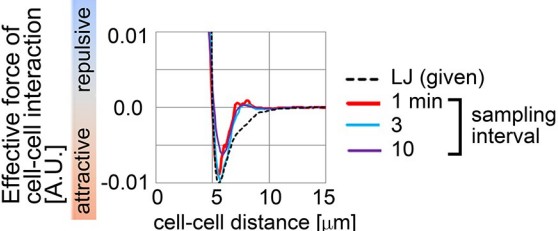

**Fig 2. Validation of inference method using simulation data. Simulations were performed on the basis of a distance–force (DF) curve obtained from the Lenard–Jones potential (LJ).** The DF curve is shown in A–i. The simulation conditions contain "Assembling process" (A), "Steady state" (B), and "Cell proliferation" (C). For A–C, snapshots of the simulations are shown where the particles are of the mean size of cells (i). In A–ii, the procedure of data analyses is exemplified, where a colored heat map representing the frequencies of the data points (Frequency index (FI)). Averaged values of the forces are shown in yellow (binned average). In A–iii, B–ii, and C–ii, the resultant effective DF curves are shown with the DF curves from the LJ potential. Normalized L2 norms = $\{F_{inferred}(D) - F_{LJ}(D)\}^2 / (F_{LJ\_max})^2$ were calculated along $D$, and then the mean values along $D$ were calculated; $F_{LJ\_max}$ is the maximum attractive force in the LJ potential, and the range of $D$ was set that the absolute values of attractive forces at $D$ are ≥ $(0.1 \times |F_{LJ\_max}|)$. The normalize L2 norms are: in A–iii, 0.035 (0%), 0.034 (10%), 0.033 (100%), and 0.024 (1000%); in B–ii, 0.032 (1min), 0.046 (3min), and 0.15 (5min); in C–ii, 0.068 (1min), 0.035 (3min), and 0.095 (10min). Related figures: S2 and S3 Figs (in the case of other given potential and random walk).

particle–particle distances (Fig 2A-ii, heat map), and by calculating binned averages, we obtained distance–force (DF) curves (Fig 2A-ii, yellow). In the absence of the fluctuations, the data points were located around the DF curve (Fig 2A-ii, standard deviation (SD) = 0%), and

the DF curve showed a similar profile to that from the LJ potential (Fig 2A-iii, LJ vs. 0%). In the presence of the fluctuations (SD = 100 or 1,000%), although the data points were widely distributed (Fig 2A-ii), the resultant DF curves were equivalent to that of the LJ potential and of that under SD = 0% (Fig 2A-iii). We also quantitatively evaluated the differences of the force values between the inferred DF curves and the LJ potential by calculating L2 norm which is described in the legend; i.e., L2 norm = $\{F_{\text{inferred}}(D) - F_{\text{LJ}}(D)\}^2$, where $F_{\text{inferred}}$ and $F_{\text{LJ}}$ are the force values from the inferred DF curves and the LJ potential, respectively. These results suggest that our method is correctly applicable for assembling process even under large fluctuations of the forces. Effect of $\omega_{F_0}$ in Eq 3 is shown in S2 Fig. In addition to the LJ potential, we also tested other potentials (e.g., Morse potential) and the inferred DF curves were similar to the given potentials (S2 and S3 Figs).

In Fig 2B, effective DF curves were inferred in systems under steady states where particles continuously moved due to the force fluctuations (SD = 1,000%). The inferred DF curves showed an almost similar profile to that of the LJ potential (Fig 2B-ii, sampling interval = 1min), but increased sampling intervals caused inconsistency with the LJ potential (sampling interval = 5 min).

In Fig 2C, effective DF curves were inferred in systems with cell proliferation where cell numbers were increased from 38 to 47; handling of cell proliferation/division is described in the S1 Text with the case of dead/lost cells. The inferred DF curves showed a similar profile to that of the LJ potential (Fig 2C-ii), and the influence of the sampling interval was similar to Fig 2B.

These results suggest that our method can correctly infer the forces of cell–cell interactions in three-dimensional cell aggregates. Note that long sampling intervals cause incorrectness of the inference in practice. In addition, around longer cell–cell distances, the inferred DF curves showed slight inconsistencies with the LJ potential in the case of Fig 2B and 2C but not 2A. This may be caused by crowding, because the particles in Fig 2A were sparsely distributed compared with Fig 2B and 2C.

## Spatial constraints had no influence on effective forces

Next, we assessed the applicability of our method to systems with spatial constraints resembling the zona pellucida or eggshells, which the mouse and *C. elegans* embryos possess. The particles with the LJ potential were put into spherocylindrical constraints (Fig 3-i). In longer lengths of spherocylinders, inferred DF curves were well consistent with the DF curves from the LJ potential (Fig 3-ii; length of spherocylinder ≥ 25μm). These results indicate that DF curves are correctly inferred under spatial constraints.

By contrast, under shorter spherocylinders, the profiles of inferred DF curves were shifted leftward along the distance (Fig 3-ii; length of spherocylinder ≤ 20μm). In these cases, the particles were compressed, and their movements were significantly restricted as follows. The mean diameter of the particles roughly corresponds to the distance providing the potential minimum (Fig 1A-iii and 1A-iv). In Fig 3-i, the particle diameters were depicted to be equivalent to the mean diameter of cells, and, under the condition of the length = 16μm, the distances between the adjacent particles seemed to be less than the diameter of the particles. Moreover, the positions of the particles were not changed but fixed throughout the simulations (Fig 3-i, 16μm), and showed a well-aligned array resembling a closed-packed state (Fig 3-i, broken lines). Compressive states are observed in real tissues such as spheroids [29,36,37], whereas we do not know the existence of tissues where cell movements are absolutely restricted by spatial constraints; if such tissues existed, DF curves could be affected by the constraints.

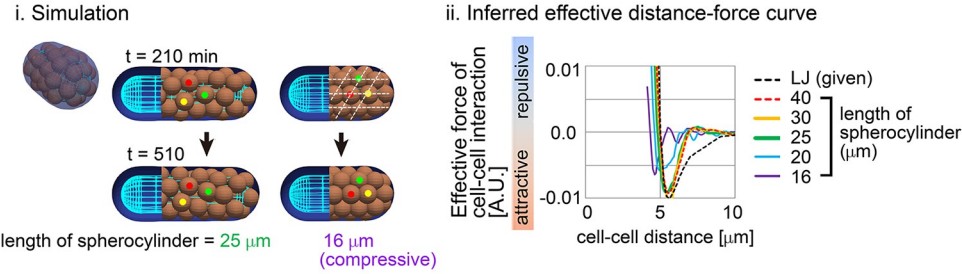

**Fig 3. Influence of external constraints to effective forces of cell–cell interaction.** Spherocylindrical constraints corresponding to eggshells are assumed in the particle model used in Fig 2 i) An example of simulations with labeling three cells (red, yellow, and green). For the condition of the spherocylindrical length = 16μm ("compressive"), the array of particles is marked by broken lines. ii) Inferred DF curves. The normalize L2 norms defined in Fig 2 are: 0.062 (40μm), 0.066 (30μm), 0.077 (25μm), 0.16 (20μm), and 0.29 (16μm). Related Figures: S4 (inference under spherical constraints) and S5 (inference in spherocylindrical constraints) Figs.

## Effective pairwise potentials were detected in C. elegans early embryo

We next investigated whether effective potential could be detected as a function of cell–cell distance in three-dimensional *in vivo* systems. The nematode *C. elegans* exhibits well-defined embryogenesis: cell movements in early embryos are almost identical among different individuals, and the time series of nuclear positions have been reported previously [31]. During early *C. elegans* embryogenesis, a fertilized egg (i.e. one cell) repeatedly undergoes cell division and cell differentiation in a stiff/undeformable eggshell, ultimately forming an ovoid embryo containing ~350 cells (Fig 4A and 4B, and S1 Movie). The cells become smaller through cell division [38], and the total volume of the embryo remains constant. Thus, the volume of each cell is reduced by a factor of 350 relative to the original egg, meaning that mean cell diameter is reduced by a factor of 7 (Fig 4B; e.g., the diameter at $t$ (time frame) = 195 is ~28% of that at $t$ = 16). Because the mean diameter of particles is approximately reflected on a DF curve (Fig 1A-iii and 1A-iv), we expected that inferred DF curves should be gradually shifted to the left side of the distance–force graph throughout embryogenesis.

Fig 4C are snapshots at different time points, that show nuclear positions with inferred effective forces where the color of the lines corresponds to the values of the forces (S2 Movie). Note that, in our inference in this system, we confirmed that this inference problem has essentially a unique solution (Fig 4D); the uniqueness of solutions is important to judge whether the inferred result is plausible in fitting problems as previously discussed [39,40]. We divided the whole embryogenesis into segments containing different time frames, and examined whether DF curves were detectable. Fig 4E is an example of the DF curve (binned average) at $t$ = 76–115 with all data points shown as a heat map. Although the data points were widely distributed similar to the previous figures (Fig 2A-ii), a clear DF curve was detected as shown by yellow. The DF curve exhibited a typical profile with regions of repulsive and attractive forces (Fig 4F, $t$ = 76–115). Similarly, we successfully detected the DF and DP (distance–potential) curves for each period (Fig 4F, $t$ = 16–55, 36–75, 76–115, 116–155, and 156–195). These curves were gradually shifted toward the left side, as we had theoretically expected, indicating that the inferred effective potentials were quantitatively consistent with the reduction in cell volume during embryogenesis. To evaluate differences in mechanical properties among the periods, we normalized both the distances and potentials by the distances at the potential minima (Fig 4G); the potentials were divided by the square of the distances at the potential minima.

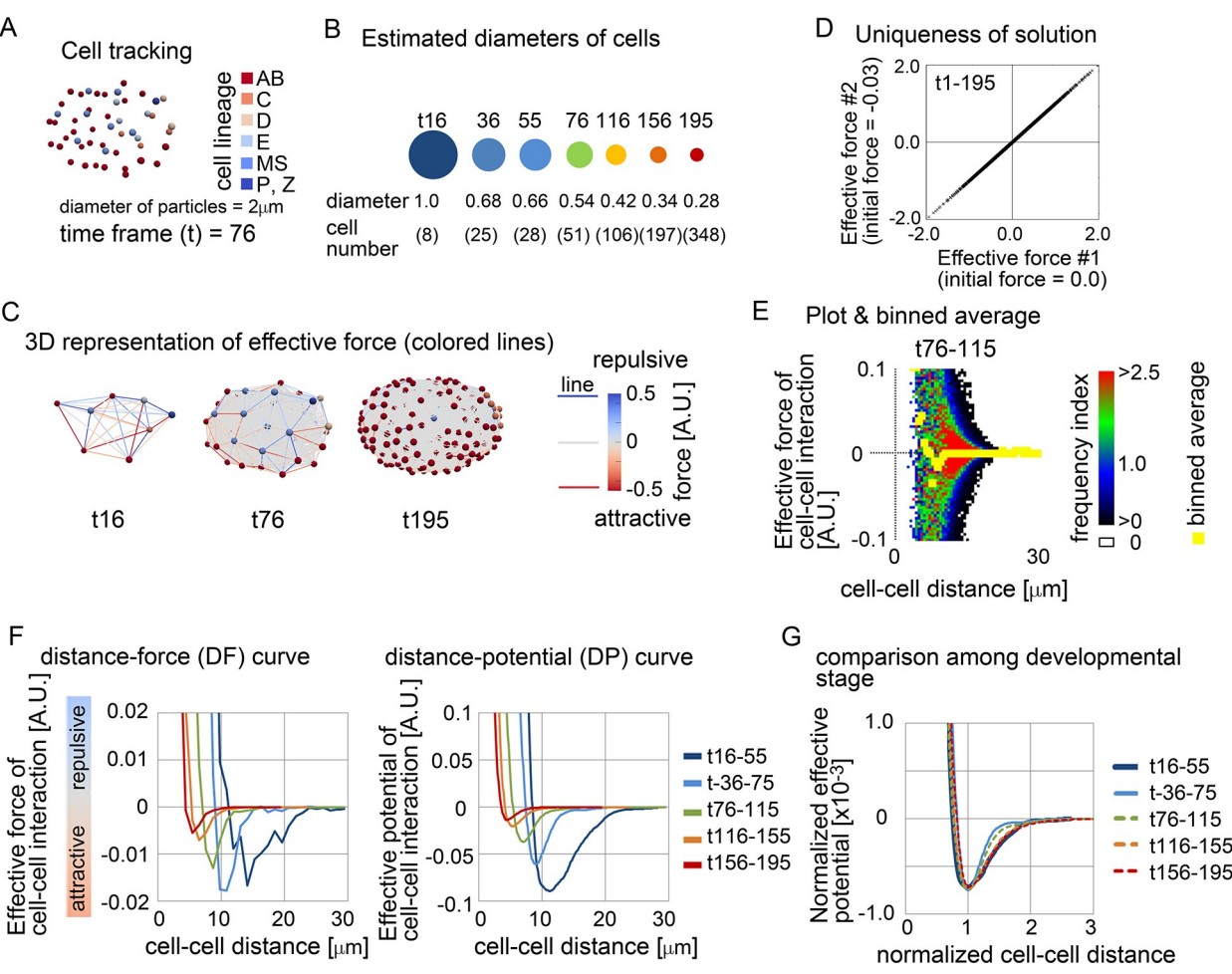

**Fig 4. Inference of effective force of cell–cell interaction in *C.elegans* embryos.** A. Snapshot of the nuclear positions in the *C.elegans* embryo with cell lineages at time frame = 76. B. Mean diameters of cells at each time frame estimated from cell numbers. Given that the volume of the embryos is constant (= *Vol*) during embryogenesis, the mean diameters were estimated from the cell numbers ($N_c$) at each time frame as follows: mean diameter $= [Vol/\{(4/3)\pi N_c\}]^{1/3}$. The diameters relative to that at time frame = 16 are shown with cell numbers. The sizes of the circles reflect the diameters, whose colors roughly correspond to the colors in the graph in F. C. Snapshots with inferred effective forces with the force values described by colored lines at $t = 16, 76$, and 195. Forces are depicted in arbitrary units (A.U.); 1 A.U. of the force can move a particle at 1μm/ min. The nuclear tracking data were obtained from a previous report [31]. D. Uniqueness of solution of effective force inference was examined. The minimizations of the cost function *G* were performed from different initial force values as described in the x–and y–axes, and the inferred values of each cell–cell interaction were plotted by crosses. E. The inferred effective forces of cell–cell interactions were plotted against the distance of cell–cell interactions with binned averages at $t = 76$–115. F. Inferred DF and DP curves at various time frames. G. DP curves normalized by the distances at the potential minima at various time frames. Related figure: S6 (uniqueness of solution was examined), S7 (the inferred DP curves were fitted by previously used frameworks such as the Morse potential), and S8 (inferred potentials under the relative velocity–based model) Figs. Related movies: S1 (tracking data) and S2 (force map) Movies.

The curves were almost unchanged during the periods, suggesting that the mechanical properties of cell–cell interactions are not drastically modulated during the development except for the cell sizes. Although slight deviations were observed for the stages t36-75 and t76-115, whether this represents changes in the mechanical properties in the embryo requires further investigation.

We also compared the profiles of the inferred DP curves with previously used ones. The framework of the Morse potential is often considered (e.g. S3 Fig) [13,18,20], while other frameworks were presented [14,16,20,29,41]. We fitted these frameworks to our inferred DP

curves. The Morse potential often fitted with small discrepancies, whereas the potential presented by Delile et al. [20] usually showed larger discrepancies (S7 Fig). This is because the Morse potential is quite flexible (i.e., various profiles are generated according to their parameter values), while the Delile's potential sets a narrower range of cell–cell distances where attractive forces can be exerted.

## Model dependency of inference in C. elegans embryo

Next, we evaluated a model-dependency. In our particle model, the viscous frictional forces were assumed to be proportional to the absolute velocities of the particles as previously presented (Eq 1). On the other hand, some papers argued that the viscous frictional forces should be determined by the velocities relative to surrounding particles [29,36]. The equation of motions in this model is as follows; $F_{C|p} = \sum_{m=1}^{N}[\{\omega_g(D_{pm})\gamma\}\{(V_{C|p} - V_{C|m}) \cdot e_{pm}\}e_{pm}]$, where $m$ is the ID of particles interacting with the $p$th particle and $V_{C|m}$ is the velocity of the $m$th particle; see the S1 Text in detail. The difference between the two models comes from whether the viscous frictional forces originate from non-moving objects (e.g., liquid medium, substrate) or moving objects (e.g., neighboring cells). S8 Fig shows the comparison of the inferred DP curves between the two models, which were almost similar profiles each other. We guess that, in systems where cell populations exhibit collective migration (i.e., similar velocities among the cells), the model choice may become significant, and, during the developmental stages in the early *C. elegans* embryo, collective migration may rarely occur.

## Effective pairwise potentials were detected in mouse pre-implantation embryos

To further investigate whether effective forces were detectable in three-dimensional real systems, we focused on mouse pre-implantation embryos, including the 8-cell and compacted morula stages (Fig 5A, illustration). In 8-cell stage embryos before compaction, cell–cell adhesion is weak, and individual cells can be easily discerned (Fig 5A, bright field). In the compaction-stage embryos composed of ~16–32 cells, cell–cell adhesion becomes stronger due to elevated expression of cadherin proteins on cell membrane, and the cells are strongly assembled [42]. The surface of the embryo becomes smooth, and the embryonic shape becomes more spherical (Fig 5A, bright field). We performed cell tracking (Fig 5A, tracking) and inferred effective forces. We successfully detected effective DF and DP curves for the two stages (Figs 5B and S9 for other embryos), which showed typical profiles with regions of repulsive and attractive forces. Compaction-stage embryos include inner and outer cells differentiating to different cell types [43]. We calculated DF curves for three pairs of interactions (inner–inner, outer–outer, and inner–outer), and found that the profiles of the curves were different each other (S10 Fig).

## Effective potentials were capable of describing morphologies

To examine whether effective pairwise potentials can be a parameter for determining morphologies, we performed simulations based on the inferred potentials from the *C. elegans* and mouse embryos. In simulations of multicellular systems, the simulation outcomes are determined by both DF curves and initial configurations, because energetic local minimum states (i.e. metastable state) as well as global minimum states are meaningful. To find stable states under the DF curves, we performed simulations starting from various initial configurations of the particles (Fig 6A, two different initial configurations are shown), and the systems were relaxed.

From the DF curves of the *C. elegans* embryos, we observed a tendency that the particles form an aggregate with an ovoid or distorted shape (Fig 6A). These results may be consistent

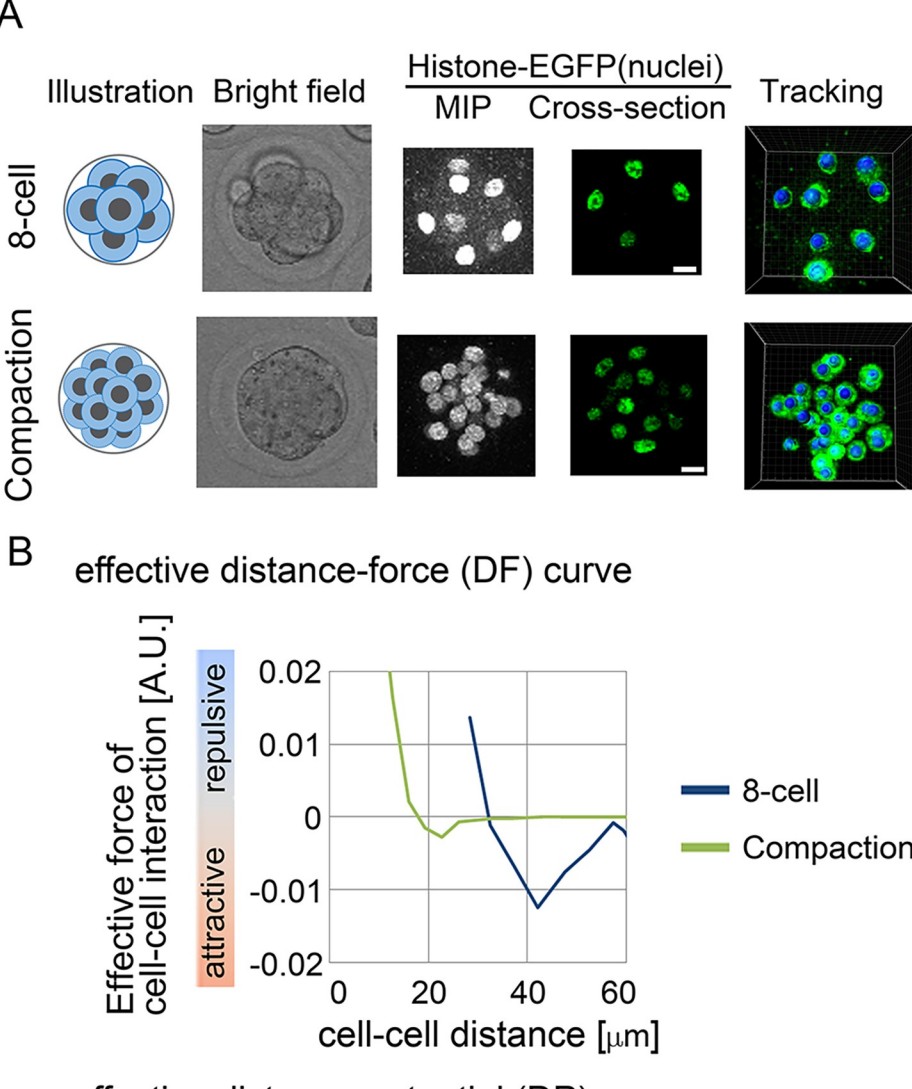

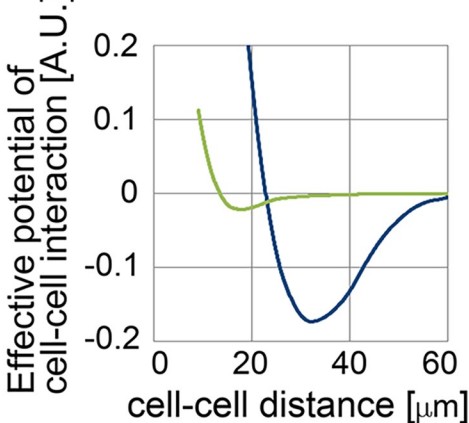

**Fig 5. Inference of the effective force of cell–cell interaction in mouse pre–implantation embryos.** A. Eight–cell and compaction stages of mouse embryo are illustrated, and their confocal microscopic images are shown: bright field, maximum intensity projection (MIP) and cross–section of fluorescence of H2B–EGFP. Snapshots of nuclear tracking

are also shown; blue spheres indicate the detected nuclei. Scale bars = 15μm. B. Inferred DF and DP curves. Related figures: S9 (data from other embryos) and S10 (DF and DP curves in the outer and inner cells in the compaction stage) Figs. Related movies: S3, S4 (tracking data), S5 and S6 (force maps) Movies.

with experimental observations: the embryonic cells can keep a cell aggregate even when the eggshell is removed, and subsequent culture leads to a cell aggregate with an ovoid or distorted shape, except for very early stage of development ($<$ ~20 cells) [44,45]. From the DF curves of the mouse 8-cell stage embryos, an aggregate was generated (Fig 6B). From the DF curves of the mouse embryos at the compaction stage, a spherical aggregate was generated (Fig 6B); this morphology is consistent with the *in vivo* situation. Therefore, the effective potentials from the *C. elegans* and mouse embryos yielded different morphologies. These results suggest that all of the DF curves are capable of recapitulating the basic morphological features of the systems.

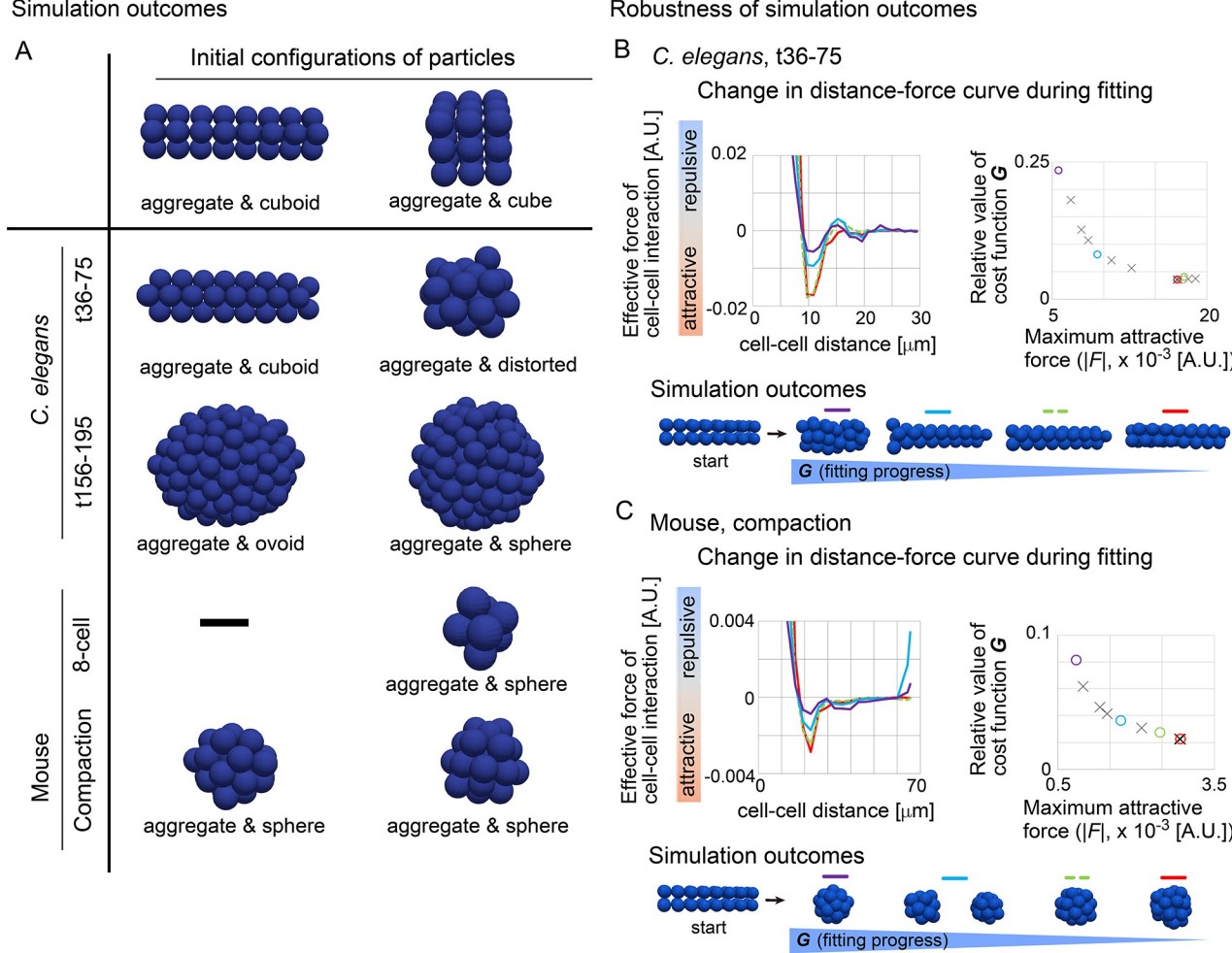

**Fig 6. Outcomes of simulations based on inferred distance–force curves.** A. Simulations results under the DF curves derived from *C. elegans* and mouse embryos are exemplified. B and C. DF curves sampled before reaching the minima of *G*, and simulation outcomes. DF curves, left panels; the values of *G* and the maximum attractive forces in the DF curves, right panels; simulation outcomes for each DF curve with the initial configuration ("start"), bottom panels. The values of *G* are relative to that in the case of all forces = 0. The colors in each three panels in B and C correspond each other. In the right panels, the numbers of the data points are 14 and 12 in B and C, respectively, and the representative ones (colored circles) are selected for presenting the DF curves and the simulation outcomes.

## Robustness of simulation outcomes

As shown above, the ovoid/distorted shape or spherical shape were generated in a manner dependent on DF curves. We examined whether our inference robustly results into the same simulation outcomes. During the minimization of the cost function $G$, we sampled DF curves before reaching the minimum of $G$. Fig 6B shows the DF curves for the *C. elegans* embryo t36-75 (left panel), and their values of $G$ are plotted against the maximum attractive force in the DF curves (right panel); the colors in the two panels correspond each other. Before reaching the minimum (a red line in left panel and a red circle in the right), the DF curves exhibited some differences in their profiles (red vs. light blue or purple). Then, simulations were performed from an initial configuration which shows a cuboid shape (bottom panel, "start"). For all the DF curves tested, the particles were not spherically assembled but the initial cuboid shape was almost sustained. By contrast, in the case of the mouse compaction stage shown in Fig 6C, the particles were spherically assembled for all the DF curves except for an intermediate sample (light blue) which shows two isolated aggregates. These results suggest that simulation outcomes resulting from our inference are robust, and the DF curves are sufficient to explain morphologies among different tissues.

## Effective pairwise potentials were different between compacted and non-compacted mouse embryo

To evaluate the predictive capability of the DF curves inferred by our method, we performed perturbation experiments. In the mouse compaction stage, the cells are tightly assembled to form a spherical and symmetric embryonic shape. In other words, we think that there are two morphological features, 1) increase in symmetry of the embryos, and 2) smoothing of the embryonic surfaces. The embryos are surrounded by external structures called zona pellucida which have negligible effects on morphogenesis [46 and reference therein]. Because both E-cadherin and actomyosin are essential for the compaction process [42,47,48], we inhibited these proteins by chemicals: EDTA (ethylenediaminetetraacetic acid) for E-cadherin, cytochalasin D for F-actin, and blebbistatin for non-muscle myosin II (Fig 7A).

The effective DP curves were inferred under these drugs (Figs 7B and S12 with enlarged views). Then, we quantitatively evaluated the differences among the DP curves, and found that the distances at the potential minima were increased under the three inhibitors (Fig 7C, left panel). Because the distances at the potential minima are related to the mean diameters of cells (Fig 1A-iv), this result predicted that the distances between adjacent cells were increased. This prediction was supported through histological observations: each cell exhibited a nearly spherical shape and seemed that the cells were not so densely assembled especially under the condition of EDTA compared with the no-drugs condition (Fig 7D). The embryonic and cellular shapes are schematically depicted in Fig 7E. In addition, we found another quantitative difference among the DP curves, related to decay of attractive forces. We measured distances at 10% energy of potential minima as relative values to the distances at the potential minima, and found that the relative distances were decreased under EDTA and blebbistatin (Fig 7C, right panel, and S14A). This result means that attractive forces of long-range interactions were reduced under these drugs.

## Effective pairwise potential described compacted and non-compacted morphologies in mouse embryo

Simulations under the DP curves from the drugs-treated embryos were performed. The cell particles were spherically or almost spherically assembled under the potentials of the normal

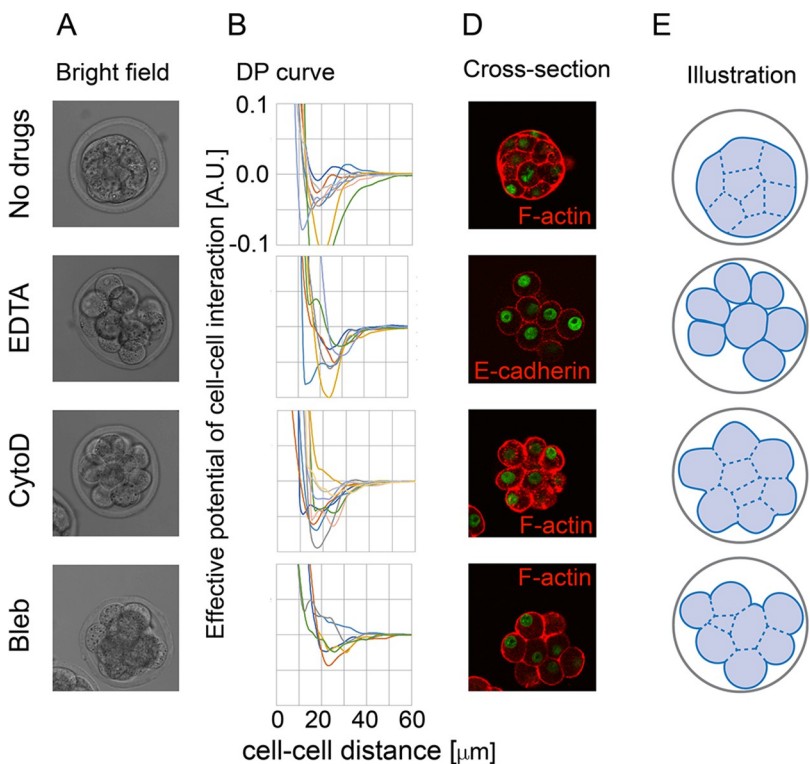

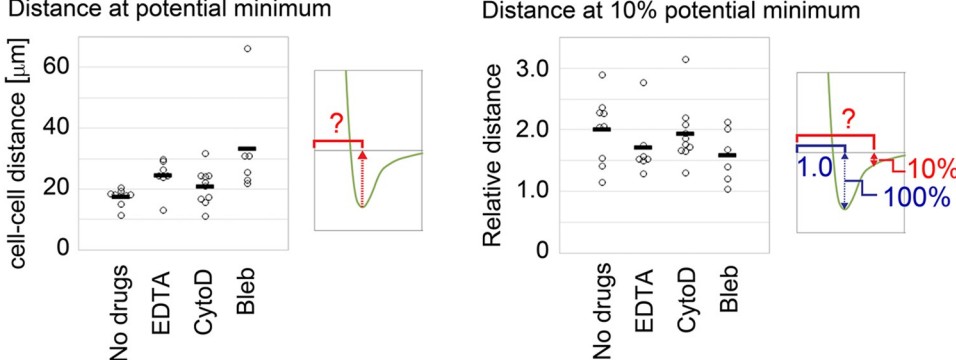

**Fig 7. Inference of effective potentials of cell–cell interaction in compaction–inhibited mouse embryos.** A. Microscopic images of embryos under chemicals. CytoD, cytochalasin D; Bleb, blebbistatin. B. Inferred DP curves in the drug–treated embryos. N = 6~10 for each condition. C. Quantitative differences in the DP curves. Distances at the potential minima in the DP curves, left panel; Distances providing 10% energy of the potential minima as the relative value to the distances at the potential minima, right panel. Mann–Whitney–Wilcoxon tests were performed and the resultant $p$–values for "No Drugs" vs. "EDTA", vs. "CytoD", and vs. "Bleb" are 0.011, 0.23, and 0.00040, respectively, in the left panel. In the right panel, distances providing other % energies instead of 10% are shown in S14B Fig. D. Confocal microscopic images of cell shapes. Cell shapes were visualized by staining F–actin or E–cadherin. E. The embryonic and cellular shapes illustrated based on D. Related figures: S11 (experimental design), S12 (enlarged view of DP curves), S13 (histological images of other embryos), and S14 (details of quantification of DP curves) Figs.

embryos or the cytochalasin D-treated embryos, respectively (Fig 8A). By contrast, the cell particles were not spherically assembled under the potentials of the EDTA or blebbistatin-treated embryos. All simulation data are provided in S15A Fig from which representative results are shown in Fig 8A.

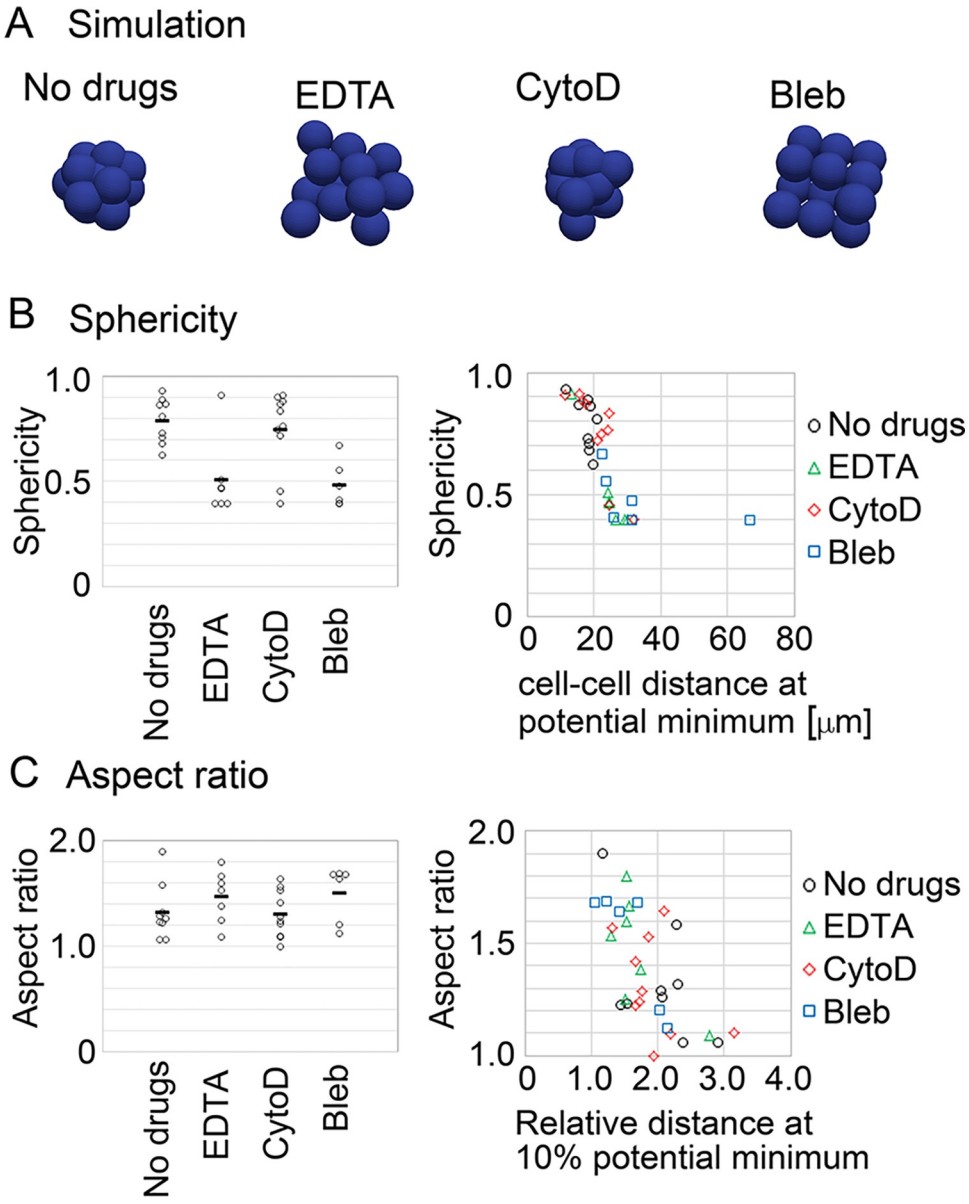

**Fig 8. Simulations under inferred distance–potential curves in drug–treated embryos, and identification of parameters explaining morphological transition.** A. Simulation results for the drug–treated embryos. All simulation data are provided in S15A Fig, and the representative results which nearly showed the mean values of the sphericities for each condition are chosen here. B. Sphericities in the simulations of the drug–treated embryos (left panel), and the sphericities plotted against the distances at the potential minima calculated in Fig 7C. Mann–Whitney–Wilcoxon tests were performed: $p$–values for "No Drugs" vs. "EDTA", vs. "CytoD", and vs. "Bleb" are 0.011, 1.0, and 0.00080, respectively. C. Aspect ratios in simulations of the drug–treated embryos (left panel), and the aspect ratios plotted against the relative distances providing 10% energy of the potential minima calculated in Fig 7C. Mann–Whitney–Wilcoxon tests were performed: $p$–values for "No Drugs" vs. "EDTA", vs. "CytoD", and vs. "Bleb" are 0.21, 0.97, and 0.33, respectively. Related figures: S15 Fig (all simulation outcomes).

To uncover what kinds of the profiles of the DP curves are linked to morphological changes in the compaction process, we first introduced two indices for the morphologies as follows. Sphericity ($= (36\pi V^2)^{1/3}/S$) is essentially defined as the ratio of volume ($V$) to surface area ($S$) so that the value becomes 1.0 for spheres. Shapes other than spheres show values less than 1.0.

Thus, both the increases in symmetry and in smoothness of surfaces contribute to this index. Aspect ratio is defined as the ratio of the length of the longest axis to that of the shortest axis in an ellipsoid which is fitted to an object of interest. The increase in symmetry leads to that the aspect ratio becomes 1.0, and otherwise the aspect ratio shows more than 1.0. The sphericities of all simulation outcomes in Figs 8A and S15A were measured, and we found that the sphericities were decreased under EDTA and blebbistatin (Fig 8B, left panel). Similarly, the aspect ratios were increased under EDTA and blebbistatin (Fig 8C, left panel).

We searched for parameters which can describe the sphericities or aspect ratios. Among several parameters tested, the distances at potential minima measured in Fig 7C showed a tight relationship to the sphericities (Fig 8B, right panel). The sphericities showed an abrupt change along the x-axis. These results indicate that the sphericity is the order parameter for the morphological transition of the compaction process and that the distance at the potential minimum is the control parameter. On the other hand, the relative distances at 10% potential minima showed a clear relationship to the aspect ratios (Fig 8C, right panel). Thus, the morphological transitions of the compaction process are induced by the quantitative changes in the DP curves.

## Discussion

In this study, we developed a top-down method for statistically inferring effective potentials of cell–cell interactions using cell tracking data. We then demonstrated that effective potentials are detectable as a function of cell–cell distances in real tissues. Our findings provide for the first time the experimental quantification of effective potentials, which have been recognized conceptually [7,8]. Furthermore, effective potentials with quantitative differences in their profiles can be a critical parameter for morphologies; ovoid vs. spherical, compacted vs. non-compacted. Moreover, the experiments using inhibitors suggest that effective potentials can be a measure for evaluating mechanical properties of cells including mutants in a noninvasive manner.

### Morphology of embryo

For the generation of spherical/circular tissues, tissue surface tension is involved in [1,29,49,50]. The surface tension reduces surface area of tissues in analogy to liquid droplets. Therefore, when non-spherical cell aggregates are provided, they show a tendency to round up to spherical shapes, where tissue surface tension contributes to the dynamics (i.e., the speed of the rounding-up) [49,51]. Although the presence of tissue surface tension may be a typical property of tissues [1,29], tissues do not always exhibit spherical shapes including the *C. elegans* embryos whose eggshells are removed [44,45]. Similarly, the drugs-treated mouse embryos showed a morphological transition from a spherical shape to a non-spherical shape after adding the drugs (S11 Fig). These data mean that tissue surface tension is not a sole determinant, but there are other factors significantly contributing to tissue shapes which can overcome the effect of the tensions. Note that cell–cell adhesion forces contribute to both tissue surface tension [26,50] and the pairwise potential of cell–cell interactions (Fig 1), suggesting that the pairwise potential relates to tissue surface tension.

In addition, the EDTA-treated mouse embryos harbor intercellular spaces (Figs 7D and S13), in which we do not think that tissue surface tensions are correctly defined. Our result identified a parameter which explains sphericity of such tissues (Fig 8B). Similar tissues with intercellular spaces are also known in other species and tissues including zebrafishes [52].

## Comparison of other inference methods with other models

To understand morphogenetic events, mechanical states and simulations based on those states are essential. Models such as vertex, Cellular Potts, and phase field models are often utilized, especially for epithelial cells [3,53–56]. These models successfully recapitulated various morphogenetic events, including cell sorting, epithelial cell movements/rearrangements, the formation of the mouse preimplantation embryos [53,57], and very early embryogenesis in *C. elegans* [58]. However, it is still challenging to measure cellular and non-cellular parameters in three-dimensional systems with high spatiotemporal resolution, although high-resolution inferences of parameters such as cell–cell junction tensions and stresses have been performed in two-dimensional situations by fitting a vertex model to experimental observations [40,59]. In addition, because the assumptions of the vertex model are based on epithelial cells, its application to other cells is limited including blastomeres and mesenchymal cells. On the other hand, particle models are mainly used for cell aggregates and mesenchymal/self-propelled cells [12,14], while some papers applied particle models to epithelial cells [15,18,19,60], meaning that the applicability of particle models is expanding. Due to its simplicity of particle models (i.e. low degrees of freedom), our method can provide a framework for quantitatively connecting model parameters to *in vivo* parameters under three-dimensional situations. Possibilities of image-based inference methods are expanding under three-dimensional situations [61].

## Limitation and perspective of our inference method

In the present study, we applied our inference method to blastomeres. Applicability of our method to other cell types including epithelial and mesenchymal cells will be evaluated. Epithelial cells are usually modeled by the vertex models where each cell is assumed to be polygonal, and many-body (i.e. $\geq 3$ cells) interactions but not pairwise ones are considered [62,63]. By using the vertex model, the highly deformable features of epithelial cells are well reproduced. It remains unknown whether pairwise potentials can be a good approximation of epithelial cells, though three-dimensional epithelial cells were modeled by particle-based models, which enabled one to simulate three-dimensionally large tissues [15,18,19,64]. In deformable objects such as cells, cell–cell interaction forces show hysteresis; i.e., the forces at the same distance become different between the cases when two cells are approaching and dissociating. This effect may be expressed by different profiles of DF curves between the two cases. In the vertex models, this hysteresis is implemented through its high deformability. Because our inference method contains temporal information of cell–cell interaction forces, it may be possible to examine whether systems show hysteresis. It would be worth investigating whether the framework of two DF curves based on the hysteresis can expand the capability of particle models for describing various phenomena.

In our inference method, the coefficient of viscous frictional forces is considered as described in Eq 1. Typically, this coefficient is assumed to be constant in multicellular models. However, the values of this coefficient and its spatiotemporal distributions in real tissues remain poorly understood [30]. There can be many factors affecting the coefficient; e.g., cell–cell contact area, cell size, the number of contacting cells per one cell, cell–cell adhesion molecules including cadherins, surface structures of cells [65–67]. In the second model presented in the *C. elegans* embryos (S8 Fig), the number of contacting cells per one cell is effectively considered. Influence of cell–cell adhesion molecules means that the coefficient is different among cell types (e.g., differentiated vs. undifferentiated cells, etc.). Unfortunately, experimental measurements of the coefficient with spatiotemporal resolution are lacking.

A serious issue to be considered is possible influence of external factors on inference results. We showed that spatial constraints such as eggshells were negligible. On the other hand, there

are many external factors in real tissues such as expanding liquid cavities in cystic structures such as the mouse blastocysts, traction forces exerted between cells and extracellular matrices (ECM) which leads to self-migratory activities, cell–ECM adhesions which leads to cell spreading, and so on [53,68,69]. It is important to evaluate whether these factors affect inferred results or not, and this issue holds true about other inference methods based on model fitting approaches (e.g. the vertex model-fitting described above). Unfortunately, it is often technically challenging to directly measure the spatiotemporal effect of the external factors on cell movements, otherwise, we can correctly infer cell–cell interaction forces by subtracting the effect of the external factors. Although overcoming this situation with many degrees of freedom is challenging, we speculate that the analogy to molecular and colloidal sciences can be helpful. In these fields, systems including external factors (e.g. solvents, pressures, etc.) can be successfully approximated as effective pairwise potentials of particle–particle interactions to some extent which are informative for predicting the systems' behaviors. We are planning to examine what kinds of external factors can be approximated as effective pairwise potentials, leading to establishment of a framework for simulating larger tissues/organs in a minimal model.

## Materials and methods

### Ethics statement

Animal care and experiments were conducted in accordance with National Institutes of Natural Sciences (NINS), the Guidelines of Animal Experimentation. The animal experiments were approved by The Institutional Animal Care and Use Committee of NINS (approval numbers; 17A030, 18A026, 19A021, 20A014, 21A039, 22A019, 23A030).

Mouse embryos were obtained after mating homozygous R26-H2B-EGFP knock-in male mice which constitutively express EGFP (enhanced green fluorescent protein)-fused H2B (histone2B proteins) [35] and ICR female mice (Japan SLC). The embryos were cultured in EmbryoMax KSOM +AA with D-Glucose (Millipore, USA) covered with mineral oil on a glass bottom dish (35-mm; 27-mm φ, Matsunami, Japan) at 37°C, and subjected to microscopic imaging. Four distinct embryos were analyzed for each embryonic stage as shown in S9 Fig.

Procedures for drug treatments are described as follows. In the case of EDTA, cytochalasin D, and blebbistatin (Figs 7 and S11), embryos were obtained at embryonic day 2.5 (E2.5), and cultured for 12–18 hours until the embryos reached ~16 cell-stage. The embryos were transferred to medium with drugs. After ~30 min culture, the compaction states of the embryos were relaxed, and live imaging was started. After the imaging, the embryos were transferred to no-drug medium (i.e. rescue), and cultured for 1–2 days. The concentrations of EDTA, cytochalasin D, and blebbistatin are 2mM, 4μg/mL, and 100μM, respectively. Under these conditions, the embryos eventually formed the blastocysts after the rescue process, meaning that the drug treatments are not lethal to the cells and embryos. Note that we used the embryos with early phase of 16 cell-stage, because the embryos with later phase were not easy to be decompacted by the drugs and application of higher drug concentrations became lethal. Six to 10 distinct embryos were analyzed for each drug condition as shown in S12 Fig.

For staining of the drug-treated embryos with anti-E-cadherin antibody or with phalloidin (Figs 7D, S13A and S13B), the embryos just before the rescue process were fixed by 4% paraformaldehyde for 1 hour under r.t.. The antibody is ECCD-2, a gift from M. Takeichi [70], and phalloidin is Alexa Fluor 594-conjugated ones (A12381; invitrogen). The secondary antibody was Alexa Fluor 594 goat anti-Rat IgG (H+L) (500×dilution) (A11007; invitrogen). For staining with FM4-64 (S13C Fig), the embryos just before the rescue process were cultured under

medium with both the drugs and FM4-64 for 30 min at 37˚C, and then imaging was performed. The concentration of FM4-64 (FM4-64FX, F34653; Invitrogen) was 5ng/μL. For three-dimensional image construction (Figs S13 and "3D"), the interval of the z-sections was 1μm.

Details of experimental procedures, mathematical modeling, and statistical methods are described in S1 Text.

## Supporting information

**S1 Text. Materials and methods for computational and experimental analyses, supplementary figures, and a list of source data.**
(DOCX)

**S1 Fig. (related to Fig 1).** Procedures to infer effective force of cell–cell interaction. A. Cut-off distance of cell–cell interaction and distance-dependent constraint for force inference. A-i. The cut-off distance is determined using the Voronoi tessellation. The Voronoi tessellation around the deepest blue particle at the center of the panel is depicted. The cut-off distance for the deepest blue particle (red broken line); particles inside the cut-off distance (deeper blue than those outside the distance). A-ii. The physically expected relationship between the distance and force. The deepest blue particle in A-i is shown. Other particles are consistent with those on the horizontal broken gray line in A-i. In regions with a distance longer than the cut-off distance, the effective forces of cell–cell interaction are 0. To achieve physically reasonable profiles of this relationship, around regions with distances slightly shorter than the cut-off distance, the values of the effective forces should be near 0 (black crosses) and not have larger values (red crosses). A-iii. During the inference of the effective forces, to avoid physically non-reasonable values of the effective forces shown in A-ii, we introduced a distance-dependent constraint $\psi(D)$ which is exponentially increased. B. Nuclear tracking data with cell division is modified to be compatible with the force inference method. During time frames ($t- \Delta t$) and $t$, a mother cell (#1) divides into two daughter cells (#2 and 3). The force inference is applied for each pair of adjacent time frames. When a cell divides, a virtual cell is assumed to exist at the centroid of the two daughter cells. Some cell–cell interactions assumed for cells #1, #2, and #3 are described by gray broken lines. The parameters in this figure are defined in S1 Text (Section 4–2).
(TIF)

**S2 Fig. (related to Fig 2).** Validation of force inference method using simulation data of particle tracking. A. Simulation data of particle tracking obtained from mathematical simulations of particles under the Lenard–Jones (LJ) or the freehand (FH) potential. The distance–force curves of these potentials are described. The dimensions of force and distance are arbitrary. The configurations of the particles at the starting and ending time points during the simulations are shown as light brown spheres. In addition, a random walk simulation is also shown. B. Snapshots of outcomes of the force inference at two time frame are exemplified. Through the force inference, effective forces are obtained for each cell–cell interaction, as shown by lines colored according to the force values (red to blue). Particles, light brown. C. Inferred effective forces from the simulations in A were plotted against the distance of cell–cell interactions. The graph space was divided into 128×128 square regions, and the frequencies of the data points plotted in each region were computed as a heat map (frequency index). Binned averages (yellow) were overlaid on the heat map (right column). The original distance–force curves defined in A are also merged (orange; given) in the case of the LJ and FH potentials. D. Effect of distance-dependent weight defined in Equation S4 on inference results. The binned average data obtained in C are used as distance–force curves. In the absence of the distance-

dependent constraint ($\alpha = 1$), the inferred distance–force curves are close to 0 (purple lines) and are far from the given profiles (gray broken lines) for both the LJ and FH potentials. When the distance-dependent weight increases, the inferred distance–force curves approach the given profiles ($\alpha = 100$, $300$). E. Sensitivity of the inference method to force scales. LJ and FH potentials were prepared with various minima of the force values, simulations were performed, and effective forces were inferred. Then, distance–force curves were estimated. The provided minima are shown in the figures (-0.03, -0.01, -0.003, and -0.001). In the case of the random walk, simulations under various magnitudes of random movements were provided as shown as relative values (1, 0.3, 0.1, 0.03, and 0.01), and the distance–force curves were randomized. (TIF)

**S3 Fig. (related to Fig 2).** Validation of force inference method for various potentials. Distance–force (DF) curves were inferred from simulation data generated under various potentials. The force values were normalized by the maximum attractive forces in the given potential. The mean cell diameters were set to be 5.0, where the forces are 0. The given potentials are the LJ (A), FH (B), and Morse (C and D) from previous two papers [18,20]. Solid lines, inferred DF curves; broken lines, the given potentials. (TIF)

**S4 Fig. (related to Fig 3).** Inference of effective forces under spherical constraints. Simulation data were used to validate our inference method. Systems with spherical constraints were considered. A. Particles were embedded into a spherical constraint (A-i and -ii). Definitions of the radius of the sphere and the radius of particles are shown (A-iii). B. The simulation procedures when a particle collides with the spherical constraint are described (dark orange circle). The detailed procedures are described in S1 Text (Section 6-2-2). C. Snapshots of simulations. The Lenard-Jones potential (LJ) was provided. Three particles are marked by red, green, and yellow. The number of particles were set to be 64 in all conditions. The condition is [sampling interval = 3.0 min, SD value of force fluctuation = 1000, and persistency of force fluctuation = 1 min]. D. Inferred DF curves under different radius of spheres. Two snapshots under the different radius are shown. In the case that 9.5μm, the particles were very closely contacted each other so that the distances between the adjacent particles seemed to be less than the diameter of the particles. Therefore, each particle was compressed. In the case that the radius was 100μm, the spherical constraint was sufficiently large so that the particles are not in contact with the surface of the constraint. The black scale bars and the gray ones correspond to the diameter of the particles and the diameter of the spherical constraint with the radius = 14.5μm. (TIF)

**S5 Fig. (related to Fig 3).** Inference of effective forces under spherocylindrical constraints. Simulation data were used to validate our inference method. Systems with spherocylindrical constraints were implemented. A. Particles were embedded into a spherocylindrical constraint (A-i and -ii). Definitions of the radius and the length of the spherocylinder, and the radius of particles are shown (A-ii and -iii). The simulation procedures when a particle collides with the spherocylindrical constraint were implemented in a similar manner to S4 Fig. B. Inferred DF curves under different radius of the spherocylindrical constraints. Two snapshots under the different radius are shown. The particles are in close contact with the surface of the constraint in the case that the radius was 7.5μm. In the case that the radius was 22.5μm, the spherocylindrical constraint was sufficiently large so that the particles are not in contact with the surface of the constraint. The number of particles were set to be 54 in all conditions. The condition is [sampling interval = 3.0 min, SD value of force fluctuation = 1000, and persistency of force fluctuation = 1 min]. (TIF)

**S6 Fig. (related to Fig 4).** Uniqueness of solutions of effective force inference in *C. elegans*. In the top left panel, the minimizations of Equation S6 were performed from different initial force values as described in the x- and y- axes. In the top right panel, the initial forces were given as uniform random numbers ranging from -0.03 to 0.03 in the y-axis. The inferred values of each cell-cell interaction were plotted by crosses. The inferred values from the different initial force values were absolutely correlated in the all cases, suggesting that a unique solution was obtained in each system. In the bottom panels, different cut-off distances were given as indicated (3.0, 4.0, 3.2, 2.8, 2.5 and 2.0-fold diameter of cell bodies), and the minimizations were performed. The inferred values were strongly correlated when the cut-off distances were greater than 2.8-fold diameter of cell bodies, suggesting that a unique solution was obtained under these conditions.
(TIF)

**S7 Fig. (related to Fig 4).** Fitting of the previously-reported potentials to inferred distance–potential (DP) curves in *C. elegans* and mouse embryos. A. The Morse potential was used for fitting. The formula of the Morse potential is:

$U(D) = U_e[\exp\{-2a(D - D_e)\} - 2\exp\{-a(D - D_e)\}]$, where $U$ is the potential energy, $D$ is the particle–particle distance, and $U_e$, $D_e$, and $a$ are the fitting parameters. Cell–cell distances were normalized by the distances providing the potential minima in the inferred potentials. Effective potential energies were normalized so that the potential minima become -1.0. t36-75 and t156-195 are from the *C. elegans* as defined in Fig 4, and 8-cell is from the mouse 8-cell stage embryo. Fitting was performed by using the solver implemented in the Excel software. B. A potential proposed by Delile et al. was used for fitting [18]. The distance–force curve was described in the Delile's paper, from which we numerically computed the DP curve for fitting.
(TIF)

**S8 Fig. (related to Fig 4).** Inferred distance–potential (DP) curves under the assumption of the relative velocity-dependent model (Eq S1-2) in *C. elegans* embryos. A. Comparison of the inferred DP curves under the two models: the absolute velocity-based and the relative velocity-based models. The effective potential energies were normalized so that the potential minima become -1.0. The time frames (e.g., t16-55, etc) were defined in Fig 4. B. Parameter dependency of inferred DP curves in *C. elegans* embryo at the t36-75 time frame. The values of $\gamma$ in Equation S1-2 were variously set. The inferred DP curve from the absolute velocity-based model (model-1) is also presented for comparison, where $\gamma$ was set 1.0. In the relative velocity-based model, because $\omega_g(D_{pm})$ in the Equation S2-1 was set to be 1.0 at $D_{pm}$ = the diameters of cell bodies, the values of $\{\omega_g(D_{pm})\gamma\}$ at $D_{pm}$ = the diameters of cell bodies are equal to $\gamma$.
(TIF)

**S9 Fig. (related to Fig 5).** Distance–force and distance–potential curves in mouse embryos. Inferred DF and DP curves in mouse 8-cell and compaction stages. Four independent embryos (#1–4) were analyzed for each stage. #1 for each stage corresponds to Fig 5.
(TIF)

**S10 Fig. (related Fig 5).** Distance–force and distance–potential curves of different cell types in mouse embryos. A. Inferred DF and DP curves of outer (blue circles) and inner (green circles) cells in mouse compaction stage. There are three possible interactions: inner–inner, outer–outer, and inner–outer cells. These data were obtained from embryo #1 in S9 Fig, compaction. The curves of all interactions (green lines) are identical to those in S9 Fig.
(TIF)

**S11 Fig. (related to Fig 7).** Experimental design for inhibiting compaction in mouse embryos. The experimental design of Fig 7 is shown. The upper panel is the experimental scheme. The lower panels are microscopic images of embryos at each step of the experimental scheme. CytoD, cytochalasin D; Bleb, blebbistatin.
(TIF)

**S12 Fig. (related to Fig 7).** Distance–potential curve in compaction-inhibited mouse embryos. The DP curves in Fig 7 are enlarged. For visualization, the DP curves were roughly categorized into three groups; unimodal (almost single potential minimum), bimodal (almost double potential minima), and others (the distance at potential minimum is very long, or a potential maximum exists). CytoD, cytochalasin D; Bleb, blebbistatin.
(TIF)

**S13 Fig. (related to Fig 7).** Histological observations of compaction-inhibited mouse embryos. A. Immuno-staining of E-cadherin. Two examples are shown for the normal or EDTA-treated embryos (#1 and #2). Conditions without the 1st anti-body is also shown as the negative control. BF, bright field; 3D, three-dimensional image. Green, Histone-EGFP. B. Phalloidin staining for F-actin. Two examples are shown for the normal, cytochalasin D-treated, or blebbistatin-treated embryos (#1 and #2). BF, bright field; 3D, three-dimensional image. Green, Histone-EGFP. CytoD. Cytochalasin D; Bleb, blebbistatin. C. FM4-64 staining for cell membrane. Two examples are shown for the normal, EDTA-treated, cytochalasin D-treated, or blebbistatin-treated embryos (#1 and #2).
(TIF)

**S14 Fig. (related to Fig 7).** Quantitative comparison of profiles of distance–potential curves in compaction-inhibited mouse embryos. A. Related to the right panel of Fig 7C. Distances at given % of potential minima were calculated as the relative distances to the distances at the potential minima. 75, 50, 25, 10, 5, 2.5, and 1% were considered. The mean values were plotted. CytoD, cytochalasin D; Bleb, blebbistatin. B. The results under 1% in A. Mann–Whitney–Wilcoxon tests were performed and the resultant $p$-values for "No Drugs" vs. "EDTA", vs. "CytoD", and vs. "Bleb" are 0.21, 0.21, and 0.036, respectively. In the case for 10% potential minima (Fig 7C, right panel), the $p$-values are 0.30, 0.60, and 0.11. Black bar, mean.
(TIF)

**S15 Fig. (related to Fig 8).** List of simulation results of compaction-inhibited mouse embryos. Simulations were performed under the all DP curves obtained from the compaction-inhibited embryos in S12 Fig. A. The initial configuration of particles was given as an asymmetric (not symmetric nor spherical) shape (initial). The simulation results were ordered according to the sphericities. The values of the sphericities and of the aspect ratios were shown. Sph, sphericity; A.R., aspect ratio. The percentages of final shapes with sphericity > 0.8 are 56 (no drugs), 14 (EDTA), 50 (cytochalasin D), or 0% (blebbistatin), respectively. The sphericities and aspect ratios are plotted in Fig 8B and 8C. B. The initial configuration of particles was given as a nearly symmetric/spherical shape (initial), and the similar analyses to A were performed. The order of the simulation results was set to be the same as that in A.
(TIF)

**S1 Movie. 3D cell movement; *C. elegans* embryo.** This movie corresponds to Fig 4A, panel "Cell tracking".
(AVI)

**S2 Movie. 3D force map of cell–cell interaction; *C. elegans* embryo.** This movie corresponds to Fig 4C, panel "3D representation of effective force".
(AVI)

**S3 Movie. Live imaging of three-dimensional systems; mouse 8-cell stage.**
(MOV)

**S4 Movie. Live imaging of three-dimensional systems; mouse compaction stage.** S3 and S4 Movies correspond to Fig 8A, panel "Tracking".
(MOV)

**S5 Movie. 3D force map of cell–cell interaction; mouse 8-cell stage.**
(AVI)

**S6 Movie. 3D force map of cell–cell interaction; mouse compaction stage.** S5 and S6 Movies correspond to Fig 5B.
(AVI)

## Acknowledgments

We thank Drs. Jean-François Joanny, Hiroaki Takagi and Yasuhiro Inoue for critical reading of the manuscript. We thank Dr. Yoshitaka Kimori for helpful discussions. We thank Ms. Azusa Kato for supporting nuclear tracking.

## Author Contributions

**Conceptualization:** Hiroshi Koyama, Hisashi Okumura, Atsushi M. Ito.

**Data curation:** Hiroshi Koyama.

**Formal analysis:** Hiroshi Koyama, Kazuyuki Nakamura.

**Funding acquisition:** Hiroshi Koyama.

**Investigation:** Hiroshi Koyama, Hisashi Okumura, Atsushi M. Ito, Kazuyuki Nakamura, Tetsuhisa Otani, Kagayaki Kato, Toshihiko Fujimori.

**Methodology:** Hiroshi Koyama, Kazuyuki Nakamura.

**Project administration:** Hiroshi Koyama.

**Resources:** Hiroshi Koyama, Toshihiko Fujimori.

**Supervision:** Toshihiko Fujimori.

**Validation:** Hiroshi Koyama.

**Visualization:** Hiroshi Koyama.

**Writing – original draft:** Hiroshi Koyama.

**Writing – review & editing:** Hiroshi Koyama, Hisashi Okumura, Atsushi M. Ito, Kazuyuki Nakamura, Tetsuhisa Otani, Kagayaki Kato, Toshihiko Fujimori.

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
