## [Decision Letter · Decision Letter 0]

14 Mar 2023

Dear Dr. Koyama,

Thank you very much for submitting your manuscript "Effective mechanical potential of cell–cell interaction explains three-dimensional morphologies during early embryogenesis" for consideration at PLOS Computational Biology.

As with all papers reviewed by the journal, your manuscript was reviewed by members of the editorial board and by several independent reviewers. In light of the reviews (below this email), we would like to invite the resubmission of a significantly-revised version that takes into account the reviewers' comments.

We cannot make any decision about publication until we have seen the revised manuscript and your response to the reviewers' comments. Your revised manuscript is also likely to be sent to reviewers for further evaluation.

Sincerely,

Philip K Maini

Academic Editor

PLOS Computational Biology

Daniel Beard

Section Editor

PLOS Computational Biology

Reviewer's Responses to Questions

**Comments to the Authors:**

Reviewer #1: Re: Review of "Effective mechanical potential of cell-cell interaction explains three-dimensional morphologies during early embryogenesis".

Summary:

------------

This study develops a top-down method to statistically infer the pairwise radial forces between cells using particle tracking data. The method's suitability is verified using synthetic data generated from a particle model, the methods reproduces the critical features of the pairwise interaction potential. Using cell tracking data from in vivo systems (mouse embryo, and C. elegans embryo) the authors demonstrate that the pairwise interaction potentials can be inferred from experimental data. Furthermore, upon simulating a particle model using the inferred interaction potentials, the simulated cell configurations quantitative macroscopic properties (e.g.\\ shape) matched those observed in the experimental systems. Biological system perturbations are considered, yielding different pairwise interaction potentials, and different simulations results, which once again matched experimental observations.

In my opinion, this study is timely, interesting, and provides a first method to address the long standing question of who force potentials for in-silico biological systems should be chosen. This paper will be of considerable interest to those simulating biological systems using force-based approaches. The paper is well written, easy to follow, and appears to contain all required pieces to reproduce and build on the work done. I have collected a few minor comments below (they verge on being nit-picky). I recommend this paper for publication.

Minor comments:

------------

Line 125: I don't think the statement that pairwise interactions between cells have never been measured is correct. There definitely are experiments that have looked at the pairwise interaction potentials between cells in atomic microscopy settings. Having said that, I agree with the authors general assertion that such profiles have not been inferred in multicellular systems

(as done in this study).

Line 213: Is the role of F_i^2 essentially to force a minimization of forces between cell pairs?

Fig 2: The inferred potentials can clearly match the potential dip. However, it appears that in all the scenarios the inferred potential increases faster to 0 after the dip. Is there a reason for this? Could this be caused by crowding? i.e. that piece of the potential cannot be inferred because due to crowding i.e. cell pairs of that separation are not observed in the data?

Line 267: I generally agree with the statement. However, there are maybe two points worth discussing or mentioning:

1. Cell volumes can change. Some cells feature aquaporins which facilitate the transport of water between the extracellular and intracellular space.

2. Many particle models (for instance those by Drasdo et. al for the growth of tumour spheroids) rely on cell compression in growth control (i.e. the transition from a growing cell to a quiescent cell). While cell volumes do not necessarily change, resolving these compressive forces is crucial to correctly model tumour spheroid growth.

Line 291: What does it mean to essentially have a unique solution?

Line 457: There are also 3D vertex models see the work by Durney and Feng.

Line 65: parameters at the cellular level

Line 70: interactions are considered.

Line 141: we develop a top-down method

Line 150 and following: On first look cells in embryos appear highly changing (in shape, position etc), maybe the authors could add a sentence to clarify the differences between epithelial, mesenchymal and blastomeres and their properties here.

Line 164: which originate from

Reviewer #2: The review is uploaded as an attachment

Reviewer #3: In their work, Koyoma and co-authors present a procedure for the inference of effective (relative) cell-cell interaction forces based on (relative) cell trajectories.

The work is based on simple idea of minimizing a cost function that compares the actual experimental displacement to a simulated displacement for a given cell-cell interaction force.

The idea itself is very useful, since both computational models and experimental characterizations can greatly benefit from a data-based and model-agnostic representation of the cell-cell interaction force. Moreover, the results are clearly presented and the data shown support the reported results.

The forces that are derived also appear realistic, showing a smooth potential with a relative long adhesive tail that is expected for interaction between adhering, strongly deformable mechanical entities (see further in point 8)

The results of the paper are straightforward and well-presented. However, I do have a few remarks that in my opinion warrant further discussion:

1. An important limitation is that forces are considered as pair-wise potentials, this means that for a given contact pair, the force is only dependent on state variables of the two interacting particles, but not the surrounding. In reality, this assumption will break down for the interaction between highly deformable bodies, in which the mechanical state is determined by the balance of surface tension and adhesion across all contacts. For example, a doublet that is laterally confined by other cells will not adhere to the same extent as an isolated single cells. For stiff particles / colloidal particles, this limitation does not play a role, since deformations are strongly localized (e.g. Hertz assumptions) and the pair-wise potential still holds. However, this is not expected for many cell types, certainly those that are typically modeled using vertex-like models. I think the authors should investigate the limitations of their approach for these highly deformable cells (preferably quantitatively) and discuss these in the manuscript.

2. In the same line of reasoning, their model assumes there are no contact-specific state variables (i.e. contact state) that play a role since the potential only takes into account the positions of the particles. In reality, contacts between cells involve bonds, which give rise to a clear hysteresis upon approach / retraction. This hysteresis is for relatively stiff particles captured in JKR-like potentials and for liquid-droplet / foam-type models will play an even bigger role. The authors should motivate their choice of neglecting this hysteresis. Effectively, it implies that the cell is always able to 'probe' its neighbourhood and react mechanically in accordance, even at a distance away from the cell (determined by the shape of the potential)

3. Another important limitation in the potential is the lack of a velocity-dependent component in the interaction force. Typically, individual cell-based models take these velocity-dependent forces into account, as relative frictional forces, see e.g. this old work:

Drasdo, Dirk, Stefan Hoehme, and Michael Block. "On the role of physics in the growth and pattern formation of multi-cellular systems: what can we learn from individual-cell based models?." Journal of Statistical Physics 128 (2007): 287-345.

In case of C. elegans, it makes sense that there are strong viscous contributions that determine the relative velocities (and thus relative displacements) of adjacent cells. For example, the cortical material of the egg has revealed a hydrodynamic length-scale of 68 um.

Saha, Arnab, et al. "Determining physical properties of the cell cortex." Biophysical journal 110.6 (2016): 1421-1429.

If this is the case, viscous forces can be propagated over long distances in the embryo and the intercellular potential will not only be determined by the relative distance between two cells, but also by their relative velocity. It would greatly benefit the generality of this paper if the authors assess the effect of such a contribution in their model. In this case, Eq. 1 would not be correct, as this equation uses a single global damping coefficient gamma (representing some kind of medium damping), rather than a viscosity that would be dependent on the relative velocity of the two cells. For C. Elegans, these relative velocities are available, so it would be a matter of including this additional parameter in their model.

4. On line 210 it is explained that the cost function Eq. 3 'worked well'. The authors should in the main text specify more clearly what it means to 'work well', and why the previous proposal does not 'work well'.

5. Small remark, but the shape in Fig. 3 is a spherocylinder or capsule, not a cylinder.

6. I do not agree with the reasoning in line 266-270. There are cells that can live in a compressed context and it is not because the cytosol itself is incompressible that cells cannot be in a general compressed state (meaning that they exert a net repulsive force on their neighhourhood). More weakly, it is true that in most tissues, a tensile state is observed (if the boundary conditions permit). However, in some cases it is obvious that local compressed states must exist: For example, take a free tissue spheroid or cell aggregate: The cells at the periphery will generate a net tissue surface tension. Young-laplace law dictates that the core of the tissue spheroid must be under pressure, and that, on average, these cells must exist in a compressed state, see e.g.

Cuvelier, M., Pešek, J., Papantoniou, I., Ramon, H., & Smeets, B. (2021). Distribution and propagation of mechanical stress in simulated structurally heterogeneous tissue spheroids. Soft Matter, 17(27), 6603-6615.

I think this reasoning should be removed from the paper as it does not further benefit the presentation of the results.

7. I am not convinced by the results presented in Fig 6 and Fig 8, which aim to show that for the measured differences in cell potential, different behaviors of cell assemblies will be expected. I think these figures are very qualitative and miss any depth or physical interpretation. If the aim is truly to investigate the difference between these potentials on tissue morphology, the rheological and physical properties of these tissues arising from the difference in potential should be investigated. E.g. the difference in tissue surface tension, shear rigidity, apparent elasticity. Also, it is expected that glassy states emerge for such sphere-approximations of cells. In practice these could be overcome by actual deformable cells and actual cell activity, which is not taken into account in this. I suggest that the authors remove these figures to the supplementary information, as they only make a weak point on the effect of these differences in potential.

8. The derived potentials (e.g. fig 4) look very interesting. However, it is a shame that the authors did not quantify the properties of these potential in more detail and do some comparison with classical models used in cell-based simulation. For example, it looks like these potentials more closely correspond to a model such as used in:

Delile, J., Herrmann, M., Peyriéras, N., & Doursat, R. (2017). A cell-based computational model of early embryogenesis coupling mechanical behaviour and gene regulation. Nature communications, 8(1), 13929.

(see supplementary figure 4 for that paper, invert the y-axis), which is commonly used in cell-based simulations these days, and it looks less similar to e.g. JKR or similar model descriptions.

Could the authors do an attempt to fit these very simple mathematical models to their derived potential and discuss eventual discrepancies?

**Have the authors made all data and (if applicable) computational code underlying the findings in their manuscript fully available?**

Reviewer #1: Yes

Reviewer #2: Yes

Reviewer #3: Yes

PLOS authors have the option to publish the peer review history of their article (what does this mean?). If published, this will include your full peer review and any attached files.

Reviewer #1: No

Reviewer #2: No

Reviewer #3: **Yes: **Bart Smeets
---

## [Decision Letter · Decision Letter 1]

26 Jun 2023

Dear Dr. Koyama,

We are pleased to inform you that your manuscript 'Effective mechanical potential of cell–cell interaction explains three-dimensional morphologies during early embryogenesis' has been provisionally accepted for publication in PLOS Computational Biology.

Best regards,

Philip K Maini

Academic Editor

PLOS Computational Biology

Daniel Beard

Section Editor

PLOS Computational Biology

Reviewer's Responses to Questions

**Comments to the Authors:**

Reviewer #2: The authors have addressed the concerns raised by this reviewer. It is therefore the recommendation of this reviewer that the journal accepts the paper for publication.

Reviewer #3: I thank the authors for their extensive reply to the raised concerns and for the additional simulations and experiments that were performed to improve the manuscript.

I apologize for my misunderstanding of (my) point 2. Indeed, the temporal resolution of intercellular forces ensures that possible hysteresis effects would be automatically retrieved by this method, a possible avenue of future research

All other concerns were addressed in-depth by the authors and I think the manuscript has been significantly improved.

I think this is solid research and a great contribution to the field.

**Have the authors made all data and (if applicable) computational code underlying the findings in their manuscript fully available?**

Reviewer #2: Yes

Reviewer #3: Yes

PLOS authors have the option to publish the peer review history of their article (what does this mean?). If published, this will include your full peer review and any attached files.

Reviewer #2: No

Reviewer #3: No

---

## [Editor Report · Acceptance letter]

31 Jul 2023

PCOMPBIOL-D-22-01824R1 

Effective mechanical potential of cell–cell interaction explains three-dimensional morphologies during early embryogenesis

Dear Dr Koyama,

I am pleased to inform you that your manuscript has been formally accepted for publication in PLOS Computational Biology. Your manuscript is now with our production department and you will be notified of the publication date in due course.

With kind regards,

Lilla Horvath
